# Walk Score^®^ and Its Associations with Older Adults’ Health Behaviors and Outcomes

**DOI:** 10.3390/ijerph16040622

**Published:** 2019-02-20

**Authors:** Yung Liao, Chien-Yu Lin, Ting-Fu Lai, Yen-Ju Chen, Bohyeon Kim, Jong-Hwan Park

**Affiliations:** 1Department of Health Promotion and Health Education, National Taiwan Normal University, 162, Heping East Road Section 1, Taipei 106, Taiwan; liaoyung@ntnu.edu.tw (Y.L.); ted971345@gmail.com (T.-F.L.); 2Institute of Health Behaviors and Community Sciences, National Taiwan University, 17, Xuzhou Road, Taipei 100, Taiwan; chienyulin@ntu.edu.tw; 3Graduate Institute of Sport, Leisure and Hospitality Management, National Taiwan Normal University, 162, Heping East Road Section 1, Taipei 106, Taiwan; lulu0126@gmail.com; 4Health Behaviors & Disease Prevention Research Group, Institute of Convergence Bio-Health, Dong-A University, Busan 49201, Korea; gus8179@gmail.com

**Keywords:** walkability, neighborhood, older adult, chronic diseases

## Abstract

This study aimed to investigate the associations between Walk Score^®^ and lifestyle behaviors and health outcomes in older Taiwanese adults. A nationwide survey was conducted through telephone-based interviews with older adults (65 years and older) in Taiwan. Data on Walk Score^®^, lifestyle behaviors (physical activity, sedentary behavior, healthy eating behavior, alcohol use, and smoking status), health outcomes (overweight/obesity, hypertension, type 2 diabetes, and cardiovascular disease), and personal characteristics were obtained from 1052 respondents. A binary logistic regression adjusting for potential confounders was employed. None of the Walk Score^®^ categories were related to the recommended levels of total physical activity. The categories “very walkable” and “walker’s paradise” were positively related to total sedentary time and TV viewing among older adults. No significant associations were found between Walk Score^®^ and other lifestyle health behaviors or health outcomes. While Walk Score^®^ was not associated with recommended levels of physical activity, it was positively related to prolonged sedentary time in the context of a non-Western country. The different associations between the walk score and health lifestyle behaviors and health outcomes in different contexts should be noted.

## 1. Introduction

There is overwhelming evidence of the role of the neighborhood environment in individuals’ lifestyle behaviors and health outcomes [1,2]. Compared to psychosocial intervention, manipulation of the built environment is expected to have long-term effects on various populations [3,4]. The neighborhood built environment is particularly important for older populations because they tend to spend most of their time in it as their mobility gradually declines with age [5]. Walkability, a key concept of the built environment, is the capacity of a neighborhood to support individuals’ lifestyle behaviors such as walking and physical activity [6]. Previous studies have assessed neighborhood walkability using various measurements such as audits of streetscapes, residents’ perceptions as well as indices of land use mix diversity, street connectivity, and residential density [7,8]. Research has associated an increased duration of physical activity with a decreased risk of negative health outcomes such as being overweight [9], symptoms of depression [10,11], and overall mortality [12,13]. A deeper understanding of neighborhood walkability and older adults’ health is needed to develop initiatives and inform policymakers and urban designers on redesigning cities and suburbs to improve public health. 

Walk Score^®^ is a free, web-based, and publicly available estimate of neighborhood walkability that can minimize the limitations of observational, self-reported, and geographic information system (GIS) measures [14]. Although Walk Score^®^ was initially developed as an indicator for housing prices and environment friendly neighborhoods (i.e. walkability and transportation), previous studies showed a positive association between walk score and walking behavior and overall physical activity [15,16,17] and a negative association with sedentary behaviors such as driving a car [18]. Moreover, the walk score is negatively related to health outcomes such as risk of obesity [19] and cardiovascular diseases (CVD) [20,21], and positively related to risks of type 2 diabetes [22]. However, studies using Walk Score^®^ mostly reported data from Western countries, and the situation remains unclear in the context of non-Western countries. Thus, to our knowledge, only two studies have investigated the relationships between Walk Score^®^ and physical activity, sedentary behavior, [18] and weight status [23] in Japan, an Asian country. Compared to Western countries, differences in the context of Asian countries include high population density, long working hours, transportation mode (i.e. motorcycles), and traditionally mixed land-use [24,25]. Based on different cultural, economic, and environmental contexts, the walk score may have different effects on public health in Asian countries. In addition, since neighborhood physical attributes influence lifestyle behaviors that contribute to chronic diseases [26], it remains unclear whether the walk score is linked with other lifestyle behaviors such as current smoking status, alcohol use, and healthy eating behavior (potentially influenced by access to amenities in the neighborhood), and other health outcomes (i.e., hypertension) in older adults. To fill this research gap, the present study aimed to explore the relationships between the walk score and lifestyle behaviors (total physical activity, sedentary time, smoking behavior, alcohol use, and eating behavior) and health outcomes (risks of overweight/ obesity, hypertension, type 2 diabetes, and CVD) of Taiwanese older adults. 

## 2. Materials and Methods

### 2.1. Participants

A cross-sectional telephone-based survey was conducted across Taiwan in 2017. To ensure a representative sample, participants were randomly selected using a stratified two-stage sampling procedure. The four areas (i.e., northern, eastern, western, and southern) of Taiwan were first stratified according to geographic location. In the second stage, individuals were randomly selected based on gender and age group (aged 85+, 75–84, 65–74 years). Each interviewer underwent training and conducted a standardized questionnaire during each survey. Among 3,282 older adults asked to participate, 1,068 responded (response rate: 32.5%). After data cleaning, the data of 1052 participants was considered valid and included in our analysis (eligible rate: 32.1%). Before each telephone interview began, verbal informed consent was obtained. Furthermore, participants were not offered any rewards. The protocols of this study were assessed and approved by the Research Ethics Committee of the University (REC number: 201706HM020).

### 2.2. Outcome Variables

Seven lifestyle behaviors and four health outcomes were included in this study:
Lifestyle behaviors: (i) total physical activity, (ii) total sedentary behavior, (iii) TV viewing, (iv) driving time, (v) healthy eating behavior, (vi) alcohol use, and (vii) current smoking statusHealth outcomes: (viii) overweight/obesity, (ix) hypertension, (x) diabetes, and (xi) CVD

#### 2.2.1. Lifestyle Behaviors

##### Physical Activity

The total time spent doing physical activity was assessed using the Taiwanese version of the International Physical Activity Questionnaire (IPAQ), which is widely utilized in telephone interview surveys on older adults [27,28]. The Taiwanese version of the IPAQ has high test-retest reliability (r = 0.78) and content validity (intra-class correlation coefficient = 0.99) [29]. The self-reported time spent walking, on moderate-intensity physical activity excluding walking (e.g., dancing and table tennis), and vigorous-intensity physical activity (e.g., aerobic exercises and basketball) was summed as total physical activity. The total physical activity was then dichotomized into two categories according to the physical activity guidelines for older adults: sufficient (≥150 min/week total physical activity) and insufficient (<150 min/week total physical activity) [30]. 

##### Sedentary Behavior

The total sedentary behavior, TV viewing time, and driving time were obtained using the Measure of Older Adults’ Sedentary Time questionnaire. Total sedentary behavior in the past week was calculated as the sum of TV viewing (i.e., time spent watching TV or digital videos), screen time (i.e., time spent online via any medium such as a computer, cellphone, and tablet), reading, chatting with others while sitting, driving time, eating, sedentary hobbies (e.g., listening to music and playing cards), sitting for work or volunteering, and other sitting activities. The total sedentary behavior was then dichotomized as >8 h/d and ≤8 h/d [31]. Furthermore, two specific sedentary behaviors, namely TV viewing and driving time, were examined because they may be potentially related to environmental walkability [18,32]. The cutoffs of 2 h/d and 1 h/d were selected for TV viewing [33,34] and driving time, respectively [35]. 

##### Eating Behavior, Alcohol Use, Current Smoking Status

In addition to physical activity and sedentary behavior, eating behavior, alcohol use, and current smoking status were recorded. Based on the items used in previous studies [36], participants were asked about their healthy eating behavior (How many servings of fruits and vegetables do you consume on an average day?), total number of alcoholic drinks consumed (How many alcoholic drinks do you consume each week?), and current smoking status (Are you a current smoker?). We divided healthy eating behavior into “Yes (three servings of vegetables and two servings of fruit)” and “No” according to the Taiwanese dietary guidelines [37], and categorized “alcohol use” and “current smoking status” as “Yes” and “No.” 

#### 2.2.2. Health Outcomes

##### Overweight/Obesity, Hypertension, Type 2 Diabetes, CVD

Health outcomes included self-reported body mass index (BMI), hypertension, type 2 diabetes, and CVD. BMI was calculated based on self-reported height and weight and classified as “normal weight, <24 kg/m^2^” and “overweight/obesity, ≥24 kg/m^2^” according to the criteria for Asian cutoff points [38]. According to the Taiwanese Chronic Disease Survey [39], the status of hypertension, type 2 diabetes, and CVD is determined by an affirmative response to a question (yes or no). The question is as follows: “Has a doctor, nurse, or other health professional ever told you that you have hypertension (type 2 diabetes, and CVD type 2 diabetes), or do you use medications for such conditions?”

### 2.3. Exposure Variable

The exposure variable was neighborhood walkability, which was measured using the Walk Score^®^ website (Link: www.walkscore.com). Walk Score^®^ was recently confirmed as a valid measure for assessing neighborhood walkability in the Asian context [40]. Our study also found significant positive correlations between Walk Score^®^ and GIS-derived environmental attributes relevant to walking, namely residential density (*r* = 0.64), intersection density (*r* = 0.70), number of local destinations (*r* = 0.70), sidewalk availability (*r* = 0.38), and access to public transportation (*r* = 0.53) in Taiwan. Walk Score^®^ is first calculated by determining a raw score for each geographic location based on the network distance to nine amenity categories of walking destinations, namely grocery stores, restaurants, shopping, coffee shops, bank services, schools, entertainment, bookstores, and parks. These raw scores are then normalized from 0 to 100 adjusting for the “intersection density” and “block length” around each location [41,42]. To calculate the walk score of each respondent, each respondent’s residential neighborhood was manually inputted into the Walk Score^®^ website by one researcher and the validity checked by another researcher. Recent studies suggested that there may not be a linear association between the walk score and walking [16] and obesity [19]. Therefore, according to the methodology [42], the walk scores were classified into four categories: (1) “car-dependent” (walk score: 0–49), (2) “somewhat walkable” (walk score: 50–69), (3) “very walkable” (walk score: 70–89), and (4) “walker’s paradise”(walk score: 90–100). 

### 2.4. Covariates

The covariates were several demographic variables: gender, age groups (65–74, 75–84, 85+ years), education achievement (up to a high school degree or college degree or more), occupational status (full-time job or no full-time job), marital status (married or not married), and living status (living alone or living with others). 

### 2.5. Data Analysis

Data from 1052 Taiwanese older adults who completed the survey on the study variables were analyzed. Since the outcome variables were skewed (i.e., health behaviors) or categorical (i.e., health outcomes) and the walk score category was provided, an adjusted binary logistic regression was performed to investigate the relationship between the four walk score categories and seven lifestyle behaviors and four health outcome variables. For the association between walk score and each lifestyle behavior, socio-demographic covariates including gender, age group, education level, occupational status, marital status, and living status were adjusted. For the association between the walk score and each health outcome, all health lifestyle behaviors (excluding TV viewing and driving) and the abovementioned socio-demographic variables were further adjusted. We considered the total sedentary behavior; therefore, we excluded these two types of sedentary behavior to avoid issues of multicollinearity. The odds ratio and 95% confidence interval (CIs) were calculated for each variable. Inferential statistics were performed using IBM SPSS 22.0 and the level of significance was set at *P* < 0.05.

## 3. Results

### 3.1. Participants’ Demographic Characteristics

Table 1 presents the demographic characteristics of the sample. Among the participants, the mean age was 73.0 years (standard deviation (SD) = 6.10), 50.1% were male, 64.5% were aged 65–74 years, 69.7% had a high school degree and lower, 90% did not have a full-time job, 75.6% were married, and 85.7% lived with others. Older adults who lived in car-dependent neighborhoods tended to be male, had up to a high school degree, and did not have a full-time job.

We demonstrate the association between 11 health-related characteristics and 4 walk score categories in Table 2. Older adults living in a “walker’s paradise” were more likely to have a total sedentary behavior time of more than 8 h per day (41.8%) and TV viewing time of more than 2 h per day (58.5%). No proportional differences were observed for total physical activity, driving time, healthy eating behavior, alcohol use, current smoking status, BMI, hypertension, type 2 diabetes, and CVD across the walk score categories.

### 3.2. Association Between Walk Score and Lifestyle Behaviors and Health Outcomes

The adjusted logistic regression models showed the associations between 11 health-related characteristics and 4 walk score categories (Table 3). In the unadjusted model, some covariates (i.e., gender, education achievement, and occupational status) and two sedentary behaviors (i.e., total sedentary behavior and TV viewing) were associated with the walk score category (data are not shown). After adjusting for potential covariates, the associations of the walk score categories with the two studied sedentary behaviors were slightly attenuated. Participants living in very walkable neighborhoods and walker’s paradises were more likely to have a total sedentary behavior time > 8 h/day (very walkable: OR = 1.68, 95% CI = 1.13–2.48; walker’s paradise: OR = 2.28, 95% CI = 1.62–3.21) and more TV viewing time (>2 h/day) (very walkable: OR = 1.47, 95% CI = 1.03–2.09; walker’s paradise: OR = 1.50, 95% CI = 1.10–2.05). No significant relationships were found between walk score category and total physical activity, driving time, healthy eating behavior, alcohol use, and current smoking status. Furthermore, walk score category was not related to health outcome variables including BMI, hypertension, type 2 diabetes, and CVD.

## 4. Discussion

The present research is one of the limited number of studies that explore the relationships of the walk score with seven lifestyle behaviors and four health outcomes among older adults in an Asian country. The findings revealed that the walk score is not associated with physical activity recommendations, but positively related to prolonged sedentary time and TV viewing among older adults in the context of Taiwan. These findings are inconsistent with previous findings on physical activity [16,17,43] and sedentary behavior [19,44] conducted in Western countries. The present findings may have significant implications for the inconsistency in the associations of the walk score with physical activity and sedentary behavior between Western countries and other contexts. 

Contrary to our expectations, none of the walk score categories were related to physical activity, although the categories “very walkable” and “walker’s paradise” were positively related to total sedentary time and TV viewing. Previous studies provided conflicting findings regarding the association between walkability and physical activity. Many associated a higher walk score with higher physical activity [45,46], although several others demonstrated no association between walkability and physical activity [46] or walking [47]. The lack of association between Walk Score^®^ and physical activity may reflect the nature of the participants, because regardless of the walk score, around 70–80% of older adults engage in sufficient physical activity. Consistent with previous studies using GIS-derived environmental measures, a highly walkable environment and total sedentary time [48] and screen time [49] were positively associated. To our knowledge, the previous literature does not provide much evidence of the potential mechanisms regarding the different relationships between the walk score and physical activity and sedentary time in older adults. Possibly, as highly walkable neighborhood environments in Taiwan are usually crowded and accompanied by more traffic (i.e., motorcycles) [50], older adults may tend to not go outdoors and spend more time engaging in sedentary behavior and watching TV at home. In addition, total physical activity, not context-specific walking behavior was used in the present study, which may partly explain why no significant relationships were found between walk score and physical activity. These results are important in terms of informing public health policymakers and urban designers that more highly walkable neighborhoods may not necessarily facilitate older adults’ physical activity, but could prolong their sedentary time, at least in Taiwan. 

Although previous studies associated the walk score with several health outcomes such as lowered obesity [19], type 2 diabetes [22], and CVD [20,21], no significant relationship was found in our study. Our results were consistent with those of previous studies that found no associations between the walk score and overweight [51] or excessive body weight [52]. These results could also be attributed to key lifestyle behaviors such as physical activity, driving, eating, drinking, and smoking, which are strongly associated with health outcomes [53,54,55], not with the walk score in the present study. This suggests that walkability, as measured by the walk score, may not play a direct role in older adults’ health outcomes in the context of Taiwan. Future research in different contexts should be conducted to confirm these results. 

There are several limitations in the present study that must be considered. First, because of the cross-sectional design, we were unable to determine the causality between variables. Second, we used self-reported lifestyle behaviors and health outcome measures that may be subject to recall bias. Third, self-selection (older adults’ preference to live in highly walkable neighborhoods), a potential confounder, was not accounted for in this study. Fourth, previous studies suggested that the Walk Score^®^ algorithm did not account for micro-scale characteristics that may impact walking behavior, such as sufficient light and traffic volume [56,57]. Fifth, although gender differences in the associations between neighborhood walkability and health behavior have been found [58,59], the present study has not further examined this issue. Future studies examining the gender difference in the association between walk score and health behaviors/outcomes are warranted. Furthermore, in this study, the walk score was obtained using participants’ residential neighborhood, not address, because residential address is a private and sensitive question for Taiwanese older adults. In our pilot survey, a large number of older adults refused to provide their complete address. However, Walk Score^®^ has been shown to provide a visualized and valid measurement of walkability in neighborhoods [56]. In this study, the neighborhood analyzed was a relatively small area in Taiwan (mean population is nearly 3,000) [60]. Future studies using participants’ residential address to confirm our results are needed. Finally, we were unable to obtain a representative sample because a telephone survey was conducted. It is impossible to reach older adults without a household telephone in Taiwan (estimated to be around 7.3% of households) [61]. Thus, our findings may be less germane and relevant to the general population.

In conclusion, Walk Score^®^, an indicator of neighborhood walkability, was not related to the recommended levels of physical activity in this study, but positively associated with prolonged sedentary time in Taiwan, a non-Western country. Thus, the different relationships between Walk Score^®^ and lifestyle behaviors and health outcomes in different contexts should be noted.

## Figures and Tables

**Table 1 ijerph-16-00622-t001:** Demographic characteristics of participants by walk score category (*n* = 1052).

Demographic Characteristics	Total	Walk Score Category									*P*-Value
		Car-Dependent		Somewhat Walkable			Very Walkable		Walker’s Paradise		
	*n*	%	*n*	%	*n*	%	*n*	%	*n*	%	
	1052	100%	396	37.6%	136	12.9%	197	18.7%	323	30.7%	
**Gender**											<0.001
Male	527	50.1%	232	58.6%	76	55.9%	85	43.1%	134	41.5%	
Female	525	49.9%	164	41.4%	60	44.1%	112	56.9%	189	58.5%	
**Age group**											0.363
65–74 years	679	64.5%	267	67.4%	88	64.7%	120	60.9%	204	63.2%	
75–84 years	311	29.6%	104	26.3%	37	27.2%	67	34.0%	103	31.9%	
85+ years	62	5.9%	25	6.3%	11	8.1%	10	5.1%	16	5.0%	
**Education achievement**											<0.001
Up to a high school degree	733	69.7%	319	80.6%	92	67.6%	140	71.1%	182	56.3%	
College degree or more	319	30.3%	77	19.4%	44	32.4%	57	28.9%	141	43.7%	
**Occupational status**											0.005
Full-time job	105	10.0%	56	14.1%	12	8.8%	15	7.6%	22	6.8%	
No full-time job	947	90.0%	340	85.9%	124	91.2%	182	92.4%	301	93.2%	
**Marital status**											0.381
Married	795	75.6%	307	77.5%	104	76.5%	151	76.6%	233	72.1%	
Not married	257	24.4%	89	22.5%	32	23.5%	46	23.4%	90	27.9%	
**Living status**											0.554
Living alone	150	14.3%	50	12.6%	23	16.9%	27	13.7%	50	15.5%	
Living with others	902	85.7%	346	87.4%	113	83.1%	170	86.3%	273	84.5%	

**Table 2 ijerph-16-00622-t002:** Participants’ health lifestyle behaviors and outcomes by walk score category (n = 1052).

Health-Related Characteristics	Total		Walk Score Category								*p*-Value
			Car-Dependent		Somewhat Walkable		Very Walkable		Walker’s Paradise		
	*n*	%	*n*	%	*n*	%	*n*	%	*n*	%	
**Lifestyle behaviors**											
**Total PA ***											0.251
Sufficient	834	79.3%	317	80.1%	99	72.8%	156	79.2%	262	81.1%	
Insufficient	218	20.7%	79	19.9%	37	27.2%	41	20.8%	61	18.9%	
**Total SB ^†^**											<0.001
>8h/day	326	31.0%	89	22.5%	38	27.9%	64	32.5%	135	41.8%	
≤8h/day	726	69.0%	307	77.5%	98	72.1%	133	67.5%	188	58.2%	
**TV viewing**											0.010
>2h/day	560	53.2%	191	48.2%	65	47.8%	115	58.4%	189	58.5%	
≤2h/day	492	46.8%	205	51.8%	71	52.2%	82	41.6%	134	41.5%	
**Driving time**											0.154
>1h/day	195	18.5%	82	20.7%	31	22.8%	32	16.2%	50	15.5%	
≤1h/day	857	81.5%	314	79.3%	105	77.2%	165	83.8%	273	84.5%	
**Healthy eating behavior**											0.399
Yes	864	82.1%	315	79.5%	113	83.1%	164	83.2%	272	84.2%	
No	188	17.9%	81	20.5%	23	16.9%	33	16.8%	51	15.8%	
**Alcohol use**											0.701
Yes	102	9.7%	36	9.1%	17	12.5%	19	9.6%	30	9.3%	
No	950	90.3%	360	90.9%	119	87.5%	178	90.4%	293	90.7%	
**Current smoking status**											0.359
Yes	71	6.7%	33	8.3%	9	6.6%	13	6.6%	16	5.0%	
No	981	93.3%	363	91.7%	127	93.4%	184	93.4%	307	95.0%	
**Health outcomes**											
**BMI ^‡^**											0.511
Normal	557	52.9%	200	50.5%	78	57.4%	108	54.8%	171	52.9%	
Underweight/Overweight	495	47.1%	196	49.5%	58	42.6%	89	45.2%	152	47.1%	
**Hypertension**											0.489
Yes	502	47.7%	178	44.9%	64	47.1%	96	48.7%	164	50.8%	
No	550	52.3%	218	55.1%	72	52.9%	101	51.3%	159	49.2%	
**Diabetes**											0.300
Yes	201	19.1%	83	21.0%	28	20.6%	40	20.3%	50	15.5%	
No	851	80.9%	313	79.0%	108	79.4%	157	79.7%	273	84.5%	
**CVD ^§^**											0.903
Yes	198	18.8%	78	19.7%	27	19.9%	38	19.3%	55	17.0%	
No	854	81.2%	318	80.3%	109	80.1%	159	80.7%	268	83.0%	

***** PA = physical activity; ^†^ SB = sedentary behavior; **^‡^** BMI = body mass index; **^§^** CVD = cardiovascular disease.

**Table 3 ijerph-16-00622-t003:** Adjusted odds ratios (ORs) for the association of walk score category with lifestyle behaviors and health outcomes.

Health-Related Characteristics	Walk Score Category									
	Car-Dependent	Somewhat Walkable			Very Walkable			Walker’s Paradise		
	OR * (95% CI ^†^)	OR * (95% CI ^†^)			OR * (95% CI ^†^)			OR * (95% CI ^†^)		
**Lifestyle behaviors**										
**Total PA ^‡^ (ref. Insufficient)**										
Sufficient	1.00	0.65	(0.41,	1.03)	0.93	(0.60,	1.43)	1.01	(0.68,	1.49)
**Total SB ^§^ (ref. ≤ 8h/day)**										
> 8h/day	1.00	1.28	(0.81,	2.01)	**1.68**	**(1.13,**	**2.48)**	**2.28**	**(1.62,**	**3.21)**
**TV viewing (ref. ≤ 2h/day)**										
> 2h/day	1.00	0.97	(0.65,	1.44)	**1.47**	**(1.03,**	**2.09)**	**1.50**	**(1.10,**	**2.05)**
**Driving time (ref. ≤ 1h/day)**										
> 1h/day	1.00	1.18	(0.73,	1.90)	0.83	(0.52,	1.32)	0.75	(0.50,	1.14)
**Healthy eating behavior (ref. Yes)**										
No	1.00	0.81	(0.48,	1.36)	0.86	(0.54,	1.36)	0.82	(0.54,	1.23)
**Alcohol use (ref. Yes)**										
No	1.00	0.57	(0.29,	1.09)	0.59	(0.32,	1.11)	0.66	(0.38,	1.17)
**Current smoking status (ref. Yes)**										
No	1.00	1.08	(0.48,	2.39)	0.79	(0.39,	1.61)	1.12	(0.57,	2.19)
**Health outcomes**										
**BMI ^||^ (ref. Overweight/obesity)**										
Normal	1.00	1.18	(0.79,	1.77)	1.13	(0.79,	1.61)	1.07	(0.78,	1.48)
**Hypertension (ref. Yes)**										
No	1.00	0.96	(0.64,	1.44)	0.90	(0.63,	1.28)	0.76	(0.55,	1.04)
**Diabetes (ref. Yes)**										
No	1.00	1.02	(0.63,	1.67)	1.05	(0.68,	1.61)	1.41	(0.94,	2.12)
**CVD ** (ref. Yes)**										
No	1.00	0.97	(0.59,	1.59)	1.06	(0.68,	1.65)	1.18	(0.79,	1.77)

***** ORs = odds ratios; ^†^ CI = confidence interval; **^‡^** PA = physical activity; **^§^** SB = sedentary behavior; ^||^ BMI = body mass index; ** CVD = cardiovascular disease; Lifestyle behaviors adjusted for gender, age group, education achievement, occupational status, marital status, and living status; Health outcomes adjusted for gender, age group, education achievement, occupational status, marital status, and living status, total PA, total SB, healthy eating behavior, alcohol use, and current smoking status.

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
