# Peer review of "Walk Score® and Its Associations with Older Adults’ Health Behaviors and Outcomes"

_ijerph, 2019, doi:10.3390/ijerph16040622_

Round 1

Reviewer 1 Report

Major alterations

Line 120 – In the sentence that says “It has been recently reported that Walk Score is a valid measure for assessing neighborhood walkability in the context of Asia”, you display a reference to an article entitle “Validity of Walk Score® as a measure of neighborhood walkability in Japan”. In the article’s conclusion of that same article, it says, and I quote “This study found that Walk Score is a valid measure of walkability in Japan, but a further country-specific validity check is necessary to expand its application to other countries, where obtaining detailed geographic data is a challenge.” So, and based on that article, I do not agree when you claim that Walk Score is a valid measure for the context of Asia. I recommend you to either rephrase your sentence, or use other reference that supports your argument. 

Line 121 – For the same reason mention above, although you use two references of how the walk score can be calculated, in my opinion, you should have presented those scores for Taiwan in order to validate WalkScore® in Taiwan region.

Minor alterations

Line 24, 40, 118, 119, 212, 219 – Add “®” to the name of the software “Walk score”.

Line 35 – Amend “…tend to spend the most time in…” to “…tend to spend most of their time in…”. In addition, add “their” to the following sentence “…because their mobility declines gradually with age”.

Line 36 - Remove “which” from the sentence “Walkability, a key concept of built environment, which is the capacity…”.

Line 37 – I would recommend you to invert the construction of the sentence, in other words, “A deeper understanding of neighborhood walkability and older adults’ health is needed, in order to develop initiatives of informing policymakers and urban designers for redesigning cities and suburbs to improve public.”

Line 46 - Remove “on” from the sentence “Nevertheless, studies on using Walk score®…”.

Line 47 – Please insert a comma after the word “Thus”. Moreover, remove the word “far” from the sentence and I would recommend you to rewrite the sentence as “Thus, to the best of our knowledge, only two studies…”

Line 49 – I suggest you to rephrase the sentence to “…in an Asian country, namely Japan.”

Line 147 – In table 1, you display in the section of occupational status two options: employed or unemployed, where you show that 90% are unemployed. When you claim that “…90% did not have a full-time job…” is not the same as saying unemployed. Since you did not state in your paper that unemployment would be considered if the person did not have a full-time job, in my opinion, it is not accurate to say that, since a person can work only in part-time jobs and not be unemployed.

Line 149 – The same situation described in line 147 happens here “…and not have a full-time job.” Moreover, instead of “…have high school degree and lower…” I suggest you to change to “…have up to a high school degree…”

Author Response

Responses to the Reviewer 1:

Major alterations

Query 1Line 120 – In the sentence that says “It has been recently reported that Walk Score is a valid measure for assessing neighborhood walkability in the context of Asia”, you display a reference to an article entitle “Validity of Walk Score® as a measure of neighborhood walkability in Japan”. In the article’s conclusion of that same article, it says, and I quote “This study found that Walk Score is a valid measure of walkability in Japan, but a further country-specific validity check is necessary to expand its application to other countries, where obtaining detailed geographic data is a challenge.” So, and based on that article, I do not agree when you claim that Walk Score is a valid measure for the context of Asia. I recommend you to either rephrase your sentence, or use other reference that supports your argument. 

Response 1: Thank you very much for your comments. We agreed with your concern.  We have recently tested the concurrent validity of Walk Score as a measure of neighborhood walkability in Taiwan in comparison with geographic information systems (GIS)-derived environmental attributes relevant to walking. The results showed that significant positive correlations were observed between Walk Score and GIS-derived environmental attributes relevant to walking, including residential density (r=0.64), intersection density (r=0.70), number of local destinations (r=0.70), sidewalk availability (r=0.38), and access to public transportation(r=0.53). We have added these results in the Method to support our argument (page 3, lines 128-132).

As this previous study, significant positive correlations were observed between Walk Score and GIS-derived environmental attributes relevant to walking, namely residential density (r=0.64), intersection density (r=0.70), number of local destinations (r=0.70), sidewalk availability (r=0.38), and access to public transportation (r=0.53) in Taiwan.

Query 2: Line 121 – For the same reason mention above, although you use two references of how the walk score can be calculated, in my opinion, you should have presented those scores for Taiwan in order to validate Walk Score® in Taiwan region.

Response 2: Thank you very much for your suggestions. We have added the details of how walk score can be calculated in the Method section (pages 3-4, lines 132-136).

Walk Score® is firstly calculated by determining a raw score to each geographic location based on the network distance to nine amenity categories of walking destinations, including grocery, restaurants, shopping, coffee shops, bank services, schools, entertainment, bookstores, and park. Then, these raw scores are normalized from 0 to 100 with adjustment of “intersection density” and “block length” around each location 41-42.

Minor alterations

Query 3: Line 24, 40, 118, 119, 212, 219 Add “®” to the name of the software “Walk score”.

Response 3: Thank you very much for your comments. We have revised accordingly throughout our manuscript.

Query 4: Line 35 – Amend “…tend to spend the most time in…” to “…tend to spend most of their time in…”. In addition, add “their” to the following sentence “…because their mobility declines gradually with age”

Response 4: Thank you very much for your suggestions. We have revised accordingly (page 1, line 35).

Neighborhood built environment is particularly important for older populations, because they tend to spend most of their time in their neighborhood environment; because their mobility declines gradually with age 5.

Query 5: Line 36 - Remove “which” from the sentence “Walkability, a key concept of built environment, which is the capacity…”

Response 5: Thank you very much for your suggestions. We have revised accordingly (page 1, line 36).

Walkability, a key concept of built environment, is the capacity of a neighborhood to support individual’s lifestyle behaviors such as walking and physical activity 6.

Query 6: Line 37 – I would recommend you to invert the construction of the sentence, in other words, “A deeper understanding of neighborhood walkability and older adults’ health is needed, in order to develop initiatives of informing policymakers and urban designers for redesigning cities and suburbs to improve public.”

Response 6: Thank you very much for your suggestions. We have revised accordingly (page 1, lines 41-44).

A deeper understanding of neighborhood walkability and older adults’ health is needed, in order to develop initiatives of informing policymakers and urban designers for redesigning cities and suburbs to improve public.

Query 7: Line 46 - Remove “on” from the sentence “Nevertheless, studies on using Walk score®…”

Response 7: Thank you very much for your suggestions. We have revised accordingly (page 2, line 52).

Nevertheless, studies using Walk score® mostly reported data from Western countries, and the situation remains unclear in the context of non-Western countries.

Query 8: Line 47 – Please insert a comma after the word “Thus”. Moreover, remove the word “far” from the sentence and I would recommend you to rewrite the sentence as “Thus, to the best of our knowledge, only two studies…”

Response 8: Thank you very much for your suggestions. We have revised accordingly (page 2, lines 53-54).

Thus, to the best of our knowledge, only two studies have investigated relationships of Walk score®  with physical activity and sedentary behavior 18 and with weight status 23 in an Asian country, namely Japan.

Query 9: Line 49 – I suggest you to rephrase the sentence to “…in an Asian country, namely Japan.”

Response 9: Thank you very much for your suggestions. We have revised accordingly (page 2, line 55).

Thus, to the best of our knowledge, only two studies have investigated relationships of Walk score®  with physical activity and sedentary behavior 18 and with weight status 23 in an Asian country, namely Japan.

Query 10: Line 147 – In table 1, you display in the section of occupational status two options: employed or unemployed, where you show that 90% are unemployed. When you claim that “…90% did not have a full-time job…” is not the same as saying unemployed. Since you did not state in your paper that unemployment would be considered if the person did not have a full-time job, in my opinion, it is not accurate to say that, since a person can work only in part-time jobs and not be unemployed.

Response 10: Thank you very much for your comments. We apologize for making this mistake. The correct item should be “full-time job” and “not full-time job”. We have revised the statements in the Method section and Table 1 accordingly.

The covariates were several demographic variables, specifically, gender, age (65–74, 75–84, 85+ years), education achievement (up to a high school degree or college degree or more), occupational status (full-time job or not full-time job), marital status (married or not married), and living status (living alone or living with others). (page 4 lines 145-146)

Query 11: Line 149 – The same situation described in line 147 happens here “…and not have a full-time job.” Moreover, instead of “…have high school degree and lower…” I suggest you to change to “…have up to a high school degree…”

Response 11: Thank you very much for your comments. The original description of “full-time job” would be correct. Moreover, we have revised the description of education in the Method section and Table 1 accordingly.

The covariates were several demographic variables, specifically, gender, age (65–74, 75–84, 85+ years), education achievement (up to a high school degree or college degree or more), occupational status (full-time job or not full-time job), marital status (married or not married), and living status (living alone or living with others). (page 4, line 145)

I hope that you find these adjustments satisfactory and that the revised version will be acceptable for publication in the International Journal of Environmental Research and Public Health.

Sincerely yours,

Yung Liao

Reviewer 2 Report

This is an interesting study examining the potential association between neighborhood walkability and health behaviors/outcomes in older adults in Taiwan. The paper addresses a worthwhile question that is certainly in line with the aims of the journal, but I think several additions and improvements need to be made before the manuscript is suitable for publication. 

Firstly, the introduction/literature review is extremely brief and does not offer much explanation for why there should be a relationship between walkability and health outcomes despite the fact there exists substantial literature on the subject. Furthermore, what is walkability and how has it been measured or assessed in other studies? There are certainly limitations to the Walk Score metric, which need to be discussed (in the Methods if not the lit. review). Furthermore, why would the relationships between health outcomes and walkability be different for Asian countries? These are all topics that could be explored in a more comprehensive lit. review.

In the methods section, there should be a discussion of Walk Score that explains how the index is calculated and (again) what the potential strengths and weaknesses might be. Take a look at Harvey & Aultman-Hall (2016) and Bereitschaft (2018) for starters. 

I don't think it is adequately explained why the authors did not use the raw walk scores in their logistic models. 

The authors mention that 'TV viewing' and 'driving' were not included in the models as covariates but do not explain why. Also, none of the covariates are mentioned in the results or discussion; were any of these significant in the models? Could any possibly help explain the results?

Similar to the introduction, the discussion is under-developed. The authors should provide a fuller explanation as to why their results run counter to previous findings. There is actually quite a lot of literature on the physical activity of older adults, which could be brought to bear on this discussion. I could imagine older adults, for example, more willing to take part in recreational exercise if they do not live in a hi-rise building, are closer to parks and other green space, and perhaps have fewer options for home delivery. All of these conditions and more could at least be discussed here. 

Importantly, the authors seems to contradict themselves when they state on p. 8, line 187 that "Consistent with previous studies using GIS-derived 188 environmental measures, positive associations were observed between high walkable environment and total sedentary time 36 and screen time." Isn't this finding inconsistent with previous findings? Most other studies have shown that higher walkability is associated with more physical activity and better health outcomes. The authors need to look at this carefully.

Finally, the authors state that "Walk Scores were obtained using 213 participant’s residential neighborhood but not address is another limitation of the study." Why weren't exact addresses used? I would imagine that significant error could have been introduced if walk score did not recognize some of the neighborhoods or if the locations of neighborhoods according to residents and walk score did not fully align. The authors should at least comment on the location accuracy of walk score in determining neighborhoods. 

Author Response

Responses to the Reviewer 2:

General comment:

This is an interesting study examining the potential association between neighborhood walkability and health behaviors/outcomes in older adults in Taiwan. The paper addresses a worthwhile question that is certainly in line with the aims of the journal, but I think several additions and improvements need to be made before the manuscript is suitable for publication.

Response: Thank you very much for your comments. We have revised and improved our manuscript. Please find our responses as below.

Query 1: Firstly, the introduction/literature review is extremely brief and does not offer much explanation for why there should be a relationship between walkability and health outcomes despite the fact there exists substantial literature on the subject.

Furthermore, what is walkability and how has it been measured or assessed in other studies? There are certainly limitations to the Walk Score metric, which need to be discussed (in the Methods if not the lit. review). Furthermore, why would the relationships between health outcomes and walkability be different for Asian countries? These are all topics that could be explored in a more comprehensive lit. review.

Response1: Thank you very much for your comments. First, we have added some explanation for why we hypothesized that there should be an association of walkability with health outcomes based on previous findings in the Introduction section (page 1, lines 39-44).

Research has showed that increased duration of physical activity was associated with decreased risk of negative health outcomes such as overweight 9, depressive symptoms 10-11, and overall mortality 12-13. A deeper understanding of neighborhood walkability and older adults’ health is needed, in order to develop initiatives of informing policymakers and urban designers for redesigning cities and suburbs to improve public.

Second, we have added the measurement of ‘walkability’ in other relevant studies in the Introduction section. Previous studies, regardless the measurements, calculated the walkability index based on land use mix diversity, street connectivity, and residential density (page 1, lines 38-39).

Previous studies irrespective the measurement assessed neighborhood walkability based on three indices, namely land use mix diversity, street connectivity, and residential density 7-8.

Third, we have also added the details of the limitations of Walk Score’s algorithm in our study in the limitation section (page 8, lines 243-245).

Fourth, previous studies suggested that the algorithm of Walk Score® did not account for micro-scale characteristics which may impact walking behavior such as lightning and traffic volume 59-60.

Last, we have strengthened the manuscripts for explaining the association of walkability with health outcomes may be different based on different contexts in the Introduction section (page 2, lines 56-58).

Compared to Western countries, there are some different contexts such as comprehensive public transportation systems, high population densities, and traditionally mixed-land-use characters in Asian countries 24-25.

Query 2: In the methods section, there should be a discussion of Walk Score that explains how the index is calculated and (again) what the potential strengths and weaknesses might be. Take a look at Harvey & Aultman-Hall (2016) and Bereitschaft (2018) for starters

Response 2: Thank you very much for your suggestions. First of all, we have added the details of how walk score can be calculated in the Method section (pages 3-4, lines 132-136).

Walk Score® is firstly calculated by determining a raw score to each geographic location based on the network distance to nine amenity categories of walking destinations, including grocery, restaurants, shopping, coffee shops, bank services, schools, entertainment, bookstores, and park. Then, these raw scores are normalized from 0 to 100 with adjustment of “intersection density” and “block length” around each location 41-42.

Secondly, we have cited the references you provided and also added the details of the potential strengths and weaknesses of Walk Score’s algorithm in the limitation section (page 8, lines 243-245).

Fourth, previous studies suggested that the algorithm of Walk Score® did not account for micro-scale characteristics which may impact walking behavior such as lightning and traffic volume 59-60.

Query 3: I don't think it is adequately explained why the authors did not use the raw walk scores in their logistic models

Response 3: Thank you very much for your comments. We have added the reason we used walk scores as categorical variables in logistic models in this study in the Method section. First of all, the main reason is that the outcome variable in our study were either skewed (i.e. self-reported time spent in physical activity and sedentary behavior) or categorical (i.e health outcomes). We have added this reason into statistical analysis (page 4, lines 150-151).

Since the outcome variables were skewed (i.e. health behaviors) or categorical (i.e health outcomes) and category of walk score has been provided, adjusted binary logistic regression was performed to investigate the relationship of four walk score categories with five lifestyle behaviors and five health outcome variables.

Furthermore, we also referred to several relevant studies (Chiu et al. 2015; Cole et al. 2015; Hirsch et al. 2013) for supporting using walk score as a categorical variable is appropriate for representing findings (page 4, lines 138-139).

Recent studies suggested that there may not be a linear association between Walk score and walking 16 and obesity 43; therefore, according to the walk score methodology 42, Walk scores were categorized into four categories: (1) “car-dependent” (walk score: 0–49), (2) “somewhat walkable” (walk score: 50–69), (3) “very walkable” (walk score: 70–89), and (4) “walker’s paradise”(walk score: 90–100).

<References>

1.         Chiu, M., Shah, B. R., Maclagan, L. C., Rezai, M.-R., Austin, P. C., & Tu, J. V. (2015). Walk Score® and the prevalence of utilitarian walking and obesity among Ontario adults: a cross-sectional study. Health Rep, 26(7), 3.

2.         Cole, R., Dunn, P., Hunter, I., Owen, N., & Sugiyama, T. (2015). Walk Score and Australian adults' home-based walking for transport. Health Place, 35, 60-65.

3.         Hirsch, J. A., Moore, K. A., Evenson, K. R., Rodriguez, D. A., & Diez Roux, A. V. (2013). Walk Score(R) and Transit Score(R) and walking in the multi-ethnic study of atherosclerosis. Am J Prev Med, 45(2), 158-166.

Query 4: The authors mention that 'TV viewing' and 'driving' were not included in the models as covariates but do not explain why. Also, none of the covariates are mentioned in the results or discussion; were any of these significant in the models? Could any possibly help explain the results?

Response 4: Thank you very much for your comments. We have added the explanation for ‘TV viewing’ and ‘driving’ were not included as covariates in the models which investigating the association of category of Walk score with health outcomes (page 4, lines 157-158).

We had taken into account the total sedentary behavior; therefore, we excluded these two types of sedentary behavior for multicollinearity.

We have also added to the revised manuscript relevant information on unadjusted model in the Results section. In the unadjusted model, some of these covariates (i.e. gender, education achievement, and occupational status) and two sedentary behaviors (i.e. total sedentary behavior and TV viewing) were associated with the level of Walk score (Table attached below). Furthermore, the associations were slightly attenuated after adjusting for the covariates. Therefore, our findings suggested that the associations between the level of Walk score and two sedentary behaviors were not influenced by confounding and we explained our findings based on the specific environmental attributes regardless of the gender, education achievement, and occupational status in the Discussion section.

Table. Unadjusted odds ratios (ORs) for the association of walk score with lifestyle behaviors and health outcomes

Category of Walk Score

Car-Dependent

Somewhat Walkable

Very Walkable

Walker’s Paradise

OR

OR

95% CI

OR

95% CI

OR

95% CI

Gender (ref. Female)

Male

1.00

0.90

(0.60-1.33)

0.54

(0.38-0.76)

0.50

(0.37-0.68)

Age group (ref. 85+ years)

65-74 years

1.00

0.75

(0.35-1.58)

1.12

(0.52-2.41)

1.19

(0.62-2.30)

75-84 years

1.00

0.81

(0.36-1.80)

1.61

(0.73-3.57)

1.55

(0.78-3.07)

Educational achievement (ref. College degree or more)

Up to a high school degree

1.00

0.51

(0.33-0.78)

0.59

(0.40-0.88)

0.31

(0.22-0.43)

Occupational status (ref. Not full-time job)

Full-time job

1.00

0.59

(0.31-1.13)

0.50

(0.28-0.91)

0.44

(0.27-0.74)

Marital status (ref. Not married)

Married

1.00

0.94

(0.59-1.49)

0.95

(0.63-1.43)

0.75

(0.54-1.05)

Living status (ref. Living with others)

Living alone

1.00

1.41

(0.82-2.41)

1.1

(0.67-1.82)

1.27

(0.83-1.94)

Lifestyle behaviors

Total PA‡ (ref. Insufficient)

Sufficient

1.00

0.67

(0.43-1.05)

0.95

(0.62-1.45)

1.07

(0.74-1.55)

Total SB§ (ref. 8h/day)

> 8h/day

1.00

1.48

(1.00-2.18)

1.40

(0.99-1.97)

2.36

(1.74-3.19)

TV viewing (ref. 2h/day)

> 2h/day

1.00

0.98

(0.67-1.45)

1.51

(1.07-2.13)

1.51

(1.13-2.04)

 Driving time (ref. 1h/day)

> 1h/day

1.00

1.31

(0.71-1.81)

0.74

(0.47-1.17)

0.70

(0.48-1.03)

 Healthy eating behavior (ref.   Yes)

No

1.00

0.79

(0.48-1.32)

0.78

(0.50-1.22)

0.73

(0.50-1.07)

Drinking behavior (ref. Yes)

No

1.00

0.70

(0.38-1.29)

0.94

(0.52-1.68)

0.98

(0.59-1.62)

Smoking behavior (ref. Yes)

No

1.00

1.28

(0.60-2.76)

1.29

(0.66-2.50)

1.74

(0.94-3.23)

Health outcomes

BMI|| (ref. Overweight/ obesity)

Normal

1.00

1.32

(0.89-1.95)

1.19

(0.84-1.68)

1.10

(0.82-1.48)

Hypertension (ref. Yes)

No

1.00

0.92

(0.62-1.36)

0.86

(0.61-1.21)

0.79

(0.59-1.06)

Diabetes (ref. Yes)

No

1.00

1.02

(0.63-1.66)

1.04

(0.68-1.59)

1.45

(0.98-2.13)

CVD** (ref. Yes)

No

1.00

0.99

(0.61-1.61)

1.03

(0.67-1.58)

1.20

(0.82-1.75)

In the unadjusted model, some of these covariates (i.e. gender, education achievement, and occupational status) and two sedentary behaviors (i.e. total sedentary behavior and TV viewing) were associated with the category of Walk Score (data were not shown). After adjusting for potential covariates, the associations of categories of Walk Score with two of studied sedentary behaviors were slightly attenuated. (page 7, lines 179-183)

Query 5: Similar to the introduction, the discussion is under-developed. The authors should provide a fuller explanation as to why their results run counter to previous findings. There is actually quite a lot of literature on the physical activity of older adults, which could be brought to bear on this discussion. I could imagine older adults, for example, more willing to take part in recreational exercise if they do not live in a hi-rise building, are closer to parks and other green space, and perhaps have fewer options for home delivery. All of these conditions and more could at least be discussed here.

Response 5: Thank you very much for your suggestions. We have added more detailed explanations for our inconsistent findings with previous studies (page 8, lines 209-217).

Previous studies showed conflicting findings on the association between walkability and physical activity. A large number of studies supported that higher walk score associated with higher physical activity 46-47; by contrast, several studies demonstrated no association between walkability and physical activity 48-49 or walking 49. One explanation for the lack of association between Walk Score® and physical activity involved the indoor environmental attributes. Given that individual’s physical frailty increases with age, older adults with decreased mobility tend to spend most of their time indoors 50, which Walk Score® could not take into account. Furthermore, the results may reflect the nature of the participants as regardless the level of Walk score, around 70-80% older adults engaged sufficient physical activity.

Query 6: Importantly, the authors seems to contradict themselves when they state on p. 8, line 187 that "Consistent with previous studies using GIS-derived 188 environmental measures, positive associations were observed between high walkable environment and total sedentary time 36 and screen time." Isn't this finding inconsistent with previous findings? Most other studies have shown that higher walkability is associated with more physical activity and better health outcomes. The authors need to look at this carefully

Response 6: Again, thank you very much for your comments. We have revised our manuscript and rephrased the statements clearly in the Discussion section. Our results showed inconsistent findings to previous studies investigating the association of walkability with physical activity whereas consistent findings to studies investigating the association of walkability with two types of sedentary behavior. Furthermore, a conflicting finding that there was a non-significantly association between walkability and health outcomes (page 8, lines 209-217).

Previous studies showed conflicting findings on the association between walkability and physical activity. A large number of studies supported that higher walk score associated with higher physical activity 46-47; by contrast, several studies demonstrated no association between walkability and physical activity 48-49 or walking 49. One explanation for the lack of association between Walk Score® and physical activity involved the indoor environmental attributes. Given that individual’s physical frailty increases with age, older adults with decreased mobility tend to spend most of their time indoors 50, which Walk Score® could not take into account. Furthermore, the results may reflect the nature of the participants as regardless the level of Walk score, around 70-80% older adults engaged sufficient physical activity.

Query 7: Finally, the authors state that "Walk Scores were obtained using 213 participant’s residential neighborhood but not address is another limitation of the study." Why weren't exact addresses used? I would imagine that significant error could have been introduced if walk score did not recognize some of the neighborhoods or if the locations of neighborhoods according to residents and walk score did not fully align. The authors should at least comment on the location accuracy of walk score in determining neighborhoods.

Response 7: Thank you very much for your comments. We have added the explanations for analyzing neighborhoods rather than addresses in the limitation section. For Taiwanese older adults, answering individual’s residential address is a sensitive subject which makes data prohibitively difficult to collect. In our pilot surveys, a large number of older adults have refused to exposure their residential addresses. Therefore, we could analyze the data including neighborhoods rather than addresses in the study as best we could (pages 8-9, lines 249-254).

Furthermore, Walk Score® were obtained using participant’s residential neighborhood but not address in the present study because residential address is a private and sensitive question for Taiwanese older adults. In our pilot survey, a large number of older adults refused to answer the complete address. However, Walk Score® has been shown to provide a visualized and valid measurement of walkability across neighborhoods 59, which was a relatively small area in Taiwan (mean population is nearly 3,000) 63.

In addition, regarding the location accuracy of walk score in determining neighborhoods, we have recently tested the concurrent validity of Walk Score as a measure of neighborhood walkability in Taiwan in comparison with geographic information systems (GIS)-derived environmental attributes relevant to walking. The results showed that significant positive correlations were observed between Walk Score and GIS-derived environmental attributes relevant to walking, including residential density (r=0.64), intersection density (r=0.70), number of local destinations (r=0.70), sidewalk availability (r=0.38), and access to public transportation(r=0.53). We have added these results in the Method section to support our argument (page 3, lines 128-132).

As this previous study, significant positive correlations were observed between Walk Score and GIS-derived environmental attributes relevant to walking, namely residential density (r=0.64), intersection density (r=0.70), number of local destinations (r=0.70), sidewalk availability (r=0.38), and access to public transportation (r=0.53) in Taiwan.

I hope that you find these adjustments satisfactory and that the revised version will be acceptable for publication in the International Journal of Environmental Research and Public Health.

Sincerely yours,

Yung Liao

Reviewer 3 Report

The work  Walk Score® and its associations with older adults’  health behaviors and outcomes present supposedly surprising results – walk score wasn’t found as related to reported measures of physical activity or health indices. However, previous works already pointed on imperfect relations between walk score and such indicators. More than that, walk score was developed as an indicator for housing prices/ friendly environmental neighborhoods and was not meant to predict behaviors or health. Therefore, researchers may be very careful when they link walk score to such indices and draw conclusions from these assumed links.

Having said that, I still believe that the work is important and innovative, but think the authors may give an appropriate context to their research and significantly elaborated their theoretical background and discussion. More specifically:

1.       The authors are asked to provide information about walk score, its methods, uses and objectives.

2.       The authors controlled gender differences. In this context, gender differences are very important and may be significant. Please note, for example

Kelley, E.A., Kandula, N.R., Kanaya, A.M., Yen, I.H., 2016. Neighborhood walkability and walking for transport among South Asians in the MASALA Study. Journal of Physical Activity and Health 13(5), 514-519.

Wasfi, R.A., Dasgupta, K., Eluru, N., Ross, N.A., 2016. Exposure to walkable neighbourhoods in urban areas increases utilitarian walking: longitudinal study of Canadians. Journal of Transport & Health 3(4), 440-447.

3.       The authors overlooked previous works that found similar results (meaning – rejecting the hypotheses that walk score is good for health and physical activity). For example:

Riley, D.L., Mark, A.E., Kristjansson, E., Sawada, M.C., Reid, R.D., 2013. Neighbourhood walkability and physical activity among family members of people with heart disease who participated in a randomized controlled trial of a behavioural risk reduction intervention. Health & Place 21, 148-155.

Takahashi, P.Y., Baker, M.A., Cha, S., Targonski, P.V., 2012. A cross-sectional survey of the relationship between walking, biking, and the built environment for adults aged over 70 years. Risk Management and Healthcare Policy 5, 35-41.

Xu, Y., Wen, M., Wang, F., 2015. Multilevel built environment features and individual odds of overweight and obesity in Utah. Applied Geography 60, 197-203.

Other works were just partially rejected these hypotheses, for example:

Sriram, U., LaCroix, A.Z., Barrington, W.E., Corbie-Smith, G., Garcia, L., Going, S.B., LaMonte, M.J., Manson, J.E., Sealy-Jefferson, S., Stefanick, M.L., Waring, M.E., 2016. Neighborhood walkability and adiposity in the women’s health initiative cohort. American Journal of Preventive Medicine 51(5), 722-730.

In sum, the work is interesting and important. However, its theoretical background is partial.

I wish the authors good luck in their revision.

Author Response

Responses to the Reviewer 3:

General Comments

The work Walk Score® and its associations with older adults’  health behaviors and outcomes present supposedly surprising results – walk score wasn’t found as related to reported measures of physical activity or health indices. However, previous works already pointed on imperfect relations between walk score and such indicators. More than that, walk score was developed as an indicator for housing prices/ friendly environmental neighborhoods and was not meant to predict behaviors or health. Therefore, researchers may be very careful when they link walk score to such indices and draw conclusions from these assumed links. Having said that, I still believe that the work is important and innovative, but think the authors may give an appropriate context to their research and significantly elaborated their theoretical background and discussion. More specifically:

Query 1:  The authors are asked to provide information about walk score, its methods, uses and objectives

Response 1: Thank you so much for your comments. Firstly, we have added the initial purpose of Walk Score® to provide more detailed information in the Introduction section (page 2, lines 47-48).

Although Walk Score® was initially developed as an indicator for housing prices and friendly environmental neighborhoods, previous studies have found that walk score is positively associated with walking behavior and overall physical activity 15-17 and negatively associated with sedentary behaviors such as car driving 18.

Besides, we have added the details of the algorithm of Walk Score and its concurrent validity as a measure of neighborhood walkability in Taiwan in comparison with geographic information systems (GIS)-derived environmental attributes relevant to walking in the Method section (pages 3-4, lines 128-136).

As this previous study, significant positive correlations were observed between Walk Score and GIS-derived environmental attributes relevant to walking, namely residential density (r=0.64), intersection density (r=0.70), number of local destinations (r=0.70), sidewalk availability (r=0.38), and access to public transportation (r=0.53) in Taiwan. Walk Score® is firstly calculated by determining a raw score to each geographic location based on the network distance to nine amenity categories of walking destinations, including grocery, restaurants, shopping, coffee shops, bank services, schools, entertainment, bookstores, and park. Then, these raw scores are normalized from 0 to 100 with adjustment of “intersection density” and “block length” around each location 41-42.

We have also discussed the potential strengths and weaknesses of using Walk Score® as a measurement in the limitation section (page 8, lines 243-245).

Fourth, previous studies suggested that the algorithm of Walk Score® did not account for micro-scale characteristics which may impact walking behavior such as lightning and traffic volume 59-60.

Query 2: The authors controlled gender differences. In this context, gender differences are very important and may be significant. Please note, for example

Kelley, E.A., Kandula, N.R., Kanaya, A.M., Yen, I.H., 2016. Neighborhood walkability and walking for transport among South Asians in the MASALA Study. Journal of Physical Activity and Health 13(5), 514-519.

Wasfi, R.A., Dasgupta, K., Eluru, N., Ross, N.A., 2016. Exposure to walkable neighbourhoods in urban areas increases utilitarian walking: longitudinal study of Canadians. Journal of Transport & Health 3(4), 440-447

Response 2: Thank you very much for your comments. We agree with you that gender differences could be important for our topic. However, the main purpose of this study is to preliminarily examine the associations between walk score and a series of lifestyle behaviors and health outcomes among overall Taiwanese older adults. Subgroup analyses such as gender, age and SES will be conducted in our future studies. Therefore, we have cited the two references you provided and listed gender-difference as a key limitation of this study, as well as suggested future studies to further examine this issue (page 8, lines 245-248).

Fifth, although gender-differences in the associations between neighborhood walkability and health behavior have been found 61-62, the present study has not further examined this issue by gender. Future studies examining the gender-difference in the association between walk score and health behaviors/outcomes are still warranted.

Query 3: The authors overlooked previous works that found similar results (meaning – rejecting the hypotheses that walk score is good for health and physical activity).

For example:

Riley, D.L., Mark, A.E., Kristjansson, E., Sawada, M.C., Reid, R.D., 2013. Neighbourhood walkability and physical activity among family members of people with heart disease who participated in a randomized controlled trial of a behavioural risk reduction intervention. Health & Place 21, 148-155.vv

Takahashi, P.Y., Baker, M.A., Cha, S., Targonski, P.V., 2012. A cross-sectional survey of the relationship between walking, biking, and the built environment for adults aged over 70 years. Risk Management and Healthcare Policy 5, 35-41.

Xu, Y., Wen, M., Wang, F., 2015. Multilevel built environment features and individual odds of overweight and obesity in Utah. Applied Geography 60, 197-203.

Other works were just partially rejected these hypotheses, for example:

Sriram, U., LaCroix, A.Z., Barrington, W.E., Corbie-Smith, G., Garcia, L., Going, S.B., LaMonte, M.J., Manson, J.E., Sealy-Jefferson, S., Stefanick, M.L., Waring, M.E., 2016. Neighborhood walkability and adiposity in the women’s health initiative cohort. American Journal of Preventive Medicine 51(5), 722-730.

Response 3: Thank you so much for providing these key references. We have added these key references into the Discussion section for providing more comprehensive information and discussion in this manuscript.

Previous studies showed conflicting findings on the association between walkability and physical activity. A large number of studies supported that higher walk score associated with higher physical activity 46-47; by contrast, several studies demonstrated no association between walkability and physical activity 48-49 or walking 49. (page 8, lines 209-212)

Our results were consistent with previous studies which found no associations between walk score and overweight 54 or excessive body weight 55. (page 8, lines 232-233)

Query 4: In sum, the work is interesting and important. However, its theoretical background is partial. I wish the authors good luck in their revision

Response 4: Thank you so much for your comments. We have reorganized and revised our manuscript thoroughly based on reviewers’ suggestions.

I hope that you find these adjustments satisfactory and that the revised version will be acceptable for publication in the International Journal of Environmental Research and Public Health.

Sincerely yours,

Yung Liao

Round 2

Reviewer 1 Report

The authors addressed all the issues that I raised.

Author Response

Thank you very much for your positive comments. 

Reviewer 2 Report

The paper has been improved, but there are still several issues that need to be addressed. Firstly, I noticed that there are a great deal of writing/language errors, even in the newly added and revised sections of the paper. Given how common the errors are I think this will require substantial effort to correct. Some examples of incorrect or confusing language include:

p. 1, "Previous studies irrespective the measurement..."

p. 1, "redesigning cities and suburbs to improve public." public what?

p. 2, "friendly environmental neighborhoods" what are these?

p. 3, "As this previous study, significant positive correlations" 

p. 4, "behavior for multicollinearity." Due to multicollinearity?

p. 8 "A large number of studies supported that higher walk score associated with higher physical activity"

p. 8, "which may impact walking behavior such as lightning and traffic volume." lighting? 

p. 8 "which found no associations between walk score and overweight or excessive body weight"

These are just a few examples. Simply put, a lot of the writing is quite awkward and not fluent, unfortunately, which can make it difficult to read at times. I suggest significant English editing prior to publication if that's a concern for the journal. 

The introduction has been improved, but could still use some work. There are other ways to assess walkability other than using density, land use mix, etc. Audits of streetscapes and measurements of actual and perceived walking behavior, for example, have also been used. 

The authors state that "Neighborhood built environment is particularly important for older populations" but largely negate this in their discussion when they argue that walk score may not matter as much for older adults because they are not very mobile and spend much of their time indoors. This seems to be quite contradictory. Also, why would walk score be positively related to risk of type 2 diabetes and yet negatively related to other negative health outcomes? I think this needs some explanation. 

The authors suggest that "Based on different cultural, economic, and environmental contexts; walk score may have different effects on public health in Asian countries." They mention that fact that Asian cities tend to be denser, etc. but what are some of these other cultural or economic contexts and why do they matter?

In the discussion, the authors state that "One explanation for the lack of association between Walk Score® and physical activity involved the indoor environmental attributes." I think it needs to be clarified what the authors mean by indoor environmental attributes." Also, though Walk Score cannot take into account what the inddor environments are like, one might expect neighborhood walkability (whether measured using walk score or some other method) to indeed be associated with propensity for walking in that neighborhood. I don't think the authors provide a sound enough rationale here for why neighborhood walkability would not impact older adult's walking behavior in Asia (or elsehwere). Perhaps they could bring some other research into the discussion to provide context regarding elderly walking behavior and environmental attributes. 

Finally, I might consider combining the conclusion and discussion sections since the conclusion is so short. 

Author Response

Responses to the Reviewer 2:

General comment:

The paper has been improved, but there are still several issues that need to be addressed. Firstly, I noticed that there are a great deal of writing/language errors, even in the newly added and revised sections of the paper. Given how common the errors are I think this will require substantial effort to correct. Some examples of incorrect or confusing language include:

Response: Thank you very much for your comments. We have revised and improved our manuscript. Also, we have sent this manuscript for further English editing by a professional company (English editing file as appendix). Please find our responses as below.

Query 1: p. 1, "Previous studies irrespective the measurement..."

Response1: Thank you very much for your comments. We have revised this sentence accordingly. (page 1, lines 37-39)

Previous studies using various measurements assessed neighborhood walkability such as audits of streetscapes, residents' perceptions as well as indices of land use mix diversity, street connectivity, and residential density 7-8.

Query 2: p. 1, "redesigning cities and suburbs to improve public." public what?

Response 2: Thank you very much for your comments. We apologize for our mistake. We have added the missing word. (page 1, line 43)

A deeper understanding of neighborhood walkability and older adults’ health is needed to develop initiatives and inform policymakers and urban designers on redesigning cities and suburbs to improve public health.

Query 3: p. 2, "friendly environmental neighborhoods" what are these?

Response 3: Thank you very much for your comments. We have rephrased this term and added examples to be clearer. (page 2, lines 46-47)

Although Walk Score® was initially developed as an indicator for housing prices and environment friendly neighborhoods (i.e. walkability and transportation), previous studies showed a positive association between walk score and walking behavior and overall physical activity 15-17 and a negative association with sedentary behaviors such as driving a car 18.

Query 4: p. 3, "As this previous study, significant positive correlations" 

Response 4: Thank you very much for your comments. We have revised this sentence accordingly. (page 3, lines 126-127)

Our study also found significant positive correlations between Walk Score® and GIS-derived environmental attributes relevant to walking, namely residential density (r=0.64), intersection density (r=0.70), number of local destinations (r=0.70), sidewalk availability (r=0.38), and access to public transportation (r=0.53) in Taiwan.

Query 5: p. 4, "behavior for multicollinearity." Due to multicollinearity?

Response 5: Thank you very much for your suggestions. We have revised accordingly. (page 4, lines 156-157)

We considered the total sedentary behavior; therefore, we excluded these two types of sedentary behavior to avoid issues of multicollinearity.

Query 6: p. 8 "A large number of studies supported that higher walk score associated with higher physical activity"

Response 6: Thank you very much for your comments. We have revised this sentence accordingly. (page 8, lines 211-213)

Many associated a higher walk score with higher physical activity 46-47, although several others demonstrated no association between walkability and physical activity 48-49 or walking 49.

Query 7: p. 8, "which may impact walking behavior such as lightning and traffic volume." lighting?

Response 7: Thank you very much for your comments. We have revised this sentence accordingly. (page 8, lines 241-243)

Fourth, previous studies suggested that the Walk Score® algorithm did not account for micro-scale characteristics that may impact walking behavior, such as sufficient light and traffic volume 59-60.

Query 8: p. 8 "which found no associations between walk score and overweight or excessive body weight"

Response 8: Thank you very much for your comments. We have revised this sentence accordingly. (page 8, lines 230-231)

Our results were consistent with those of previous studies that found no associations between the walk score and overweight 54 or excessive body weight 55.

Query 9: These are just a few examples. Simply put, a lot of the writing is quite awkward and not fluent, unfortunately, which can make it difficult to read at times. I suggest significant English editing prior to publication if that's a concern for the journal.

Response 9: Thank you very much for your comments. We have sent this manuscript for further English editing by a professional company. Please find the English editing file as appendix.  

Query 10: The introduction has been improved, but could still use some work. There are other ways to assess walkability other than using density, land use mix, etc. Audits of streetscapes and measurements of actual and perceived walking behavior, for example, have also been used.

Response 10: Thank you very much for your comments. We have added other ways of assessing walkability to improve our introduction. (page 1, lines 37-39)

Previous studies using various measurements assessed neighborhood walkability such as audits of streetscapes, residents' perceptions as well as indices of land use mix diversity, street connectivity, and residential density 7-8.

Query 11: The authors state that "Neighborhood built environment is particularly important for older populations" but largely negate this in their discussion when they argue that walk score may not matter as much for older adults because they are not very mobile and spend much of their time indoors. This seems to be quite contradictory. Also, why would walk score be positively related to risk of type 2 diabetes and yet negatively related to other negative health outcomes? I think this needs some explanation. 

Response 11: Thank you very much for your comments. We agree with your comments. First of all, we have deleted this contradictory explanation from our Discussion. Furthermore, we found no associations between walk score and diabetes or health outcomes (Table 3), and thus we have added several explanation why no significant relationship was found in our study. (page 8, lines 231-233)

These results could also be attributed to key lifestyle behaviors such as eating, drinking, smoking, physical activity, and driving, which are strongly associated with health outcomes 56-58, not with the walk score in the present study.

Query 12: The authors suggest that "Based on different cultural, economic, and environmental contexts; walk score may have different effects on public health in Asian countries." They mention that fact that Asian cities tend to be denser, etc. but what are some of these other cultural or economic contexts and why do they matter?

Response 12: Thank you very much for your comments. We have added several examples for the cultural (i.e. transportation mode) or economic (i.e. long working hours) contexts accordingly. (page 2, lines 55-57)

Compared to Western countries, differences in the context of Asian countries include high population density, long working hours, transportation mode (i.e. motorcycles), and traditionally mixed land-use 24-25.

Query 13: In the discussion, the authors state that "One explanation for the lack of association between Walk Score® and physical activity involved the indoor environmental attributes." I think it needs to be clarified what the authors mean by indoor environmental attributes." Also, though Walk Score cannot take into account what the indoor environments are like, one might expect neighborhood walkability (whether measured using walk score or some other method) to indeed be associated with propensity for walking in that neighborhood. I don't think the authors provide a sound enough rationale here for why neighborhood walkability would not impact older adult's walking behavior in Asia (or elsewhere). Perhaps they could bring some other research into the discussion to provide context regarding elderly walking behavior and environmental attributes.

Response 13: Again, thank you very much for your comments. We have deleted this contradictory explanation (indoor environment) and focused on the possible explanation - physical activity levels in the sample of our study. (page 8, lines 211-213)

Many associated a higher walk score with higher physical activity 46-47, although several others demonstrated no association between walkability and physical activity 48-49 or walking 49. The lack of association between Walk Score® and physical activity may reflect the nature of the participants, because regardless of the walk score, around 70–80% of older adults engage in sufficient physical activity.

Query 14: Finally, I might consider combining the conclusion and discussion sections since the conclusion is so short.

Response 14: Thank you very much for your suggestions. We have combined the conclusion and discussion section. (page 9, lines 257-260)

In conclusion, Walk score®, an indicator of neighborhood walkability, was not related to the recommended levels of physical activity in this study, but positively associated with prolonged sedentary time in Taiwan, a non-Western country. The different relationships between Walk score® and lifestyle behaviors and health outcomes in different contexts should thus be noted.

I hope that you find these adjustments satisfactory and that the revised version will be acceptable for publication in the International Journal of Environmental Research and Public Health.

Sincerely yours,

Yung Liao
